

# Unilateral hamstring foam rolling does not impair strength but the rate of force development of the contralateral muscle

Xin Ye[1], Benjamin S. Killen[2], Krista L. Zelizney[3], William M. Miller[1] and Sunggun Jeon[1]

[1] Department of Health, Exercise Science, and Recreation Management, University of Mississippi, University, MS, United States of America
[2] Department of Health Related Professions, University of Mississippi Medical Center, Jackson, MS, United States of America
[3] Nicole Wertheim College of Nursing & Health Sciences, Florida International University, Miami, FL, United States of America

## ABSTRACT

**Background**. Self-administered foam rolling (SAFR) is an effective massage technique often used in sport and rehabilitation settings to improve range of motion (ROM) without impairing the strength performance. However, the effects of unilateral SAFR on contralateral non-intervened muscle's rate of force development (RFD) are unknown. Therefore, the purpose of this investigation was to examine the acute effects of unilateral hamstrings SAFR on the contralateral limb flexibility, the isometric strength, and the RFD parameters.

**Methods**. Thirty-four subjects (21 women) completed two separate randomly sequenced experimental visits, during which the control (rested for 10 min) or ten, 30-second SAFR were performed with the dominant hamstring muscle group. Before (Pre) and after (Post) the interventions, the contralateral hip flexion passive ROM, the maximal explosive isometric strength of the contralateral knee flexors with the corresponding prime mover muscles' surface electromyographic (EMG) amplitude were measured. Separate two-way (time $\times$ intervention) repeated measures analyses of variance (ANOVAs) were used to examine the potential changes of the dependent variables.

**Results**. The SAFR significantly improved the contralateral limb ROM (Pre vs. Post: $68.3 \pm 21.0$ vs. $73.2 \pm 23.2$ degrees, $p < 0.001$; $d = 0.22$). No change was found for the contralateral isometric strength or the maximal EMG amplitude. For the RFD parameters, the percent changes of the RFDs for the first 50, 100, and 200 ms of the maximal explosive isometric contraction were $-31.2\%$, $-16.8\%$, and $-10.1\%$, respectively, following the unilateral SAFR, relative to the control condition. In addition, the decrement of the first 50-ms RFD reached statistical significance ($p = 0.007$; Cohen's $d = 0.44$).

**Conclusion**. Ten sets of 30-second unilateral hamstring SAFR improved the ROM of the non-intervened contralateral limb, but decreased its ability to generate force, especially during the early phase (e.g., 50 ms) of the maximal explosive contraction.

Corresponding author
Xin Ye, xye1@olemiss.edu

## INTRODUCTION

Foam rolling is a type of massage technique that has been extensively used both in athletic and rehabilitation settings, mainly due to its portability and ease of use. Over the last decade, the increasing number of research studies have focused on the examinations of performing self-administered foam rolling (SAFR) as a pre-exercise warmup and/or a post-exercise recovery strategy. These research studies have examined the short-term effects (*Beardsley & Skarabot, 2015*; *Cheatham et al., 2015*; *Freiwald et al., 2016*; *MacDonald et al., 2013*) and the potential long-term musculoskeletal adaptations (*Macgregor et al., 2018*) following the SAFR interventions. Briefly, SAFR consists of using one's body weight to position a specific muscle group onto a dense foam roller while moving back-and-forth to simulate soft tissue mobilization. With the undulating massage-like mechanical pressure placed upon the target muscle(s), the SAFR is an effective tool to promote soft tissue extensibility (*Barnes, 1997*; *MacDonald et al., 2013*), as well as to enhance recovery from high-intensity exercise (*D'Amico & Gillis, 2017*; *Macdonald et al., 2014*; *Pearcey et al., 2015*). More specifically, the SAFR can improve joint range of motion (ROM) (*Beardsley & Skarabot, 2015*; *Cheatham et al., 2015*; *Freiwald et al., 2016*; *Healey et al., 2014*; *Junker & Stoggl, 2015*; *MacDonald et al., 2013*; *Madoni et al., 2018*; *Monteiro et al., 2017*; *Su et al., 2017*) and increase pain pressure threshold (PPT) (*Cheatham & Baker, 2017*; *Cheatham, Stull & Kolber, 2019*; *Pearcey et al., 2015*) without necessarily impairing subsequent athletic performance (*Behara & Jacobson, 2017*; *Healey et al., 2014*; *MacDonald et al., 2013*).

In addition to the aforementioned effects that SAFR directly imposes on the local intervened muscle group, potential changes in musculoskeletal functions on the contralateral homologous or non-related heterologous (nonlocal) muscle group were also observed more recently. For example, *Killen, Zelizney & Ye (2018)* showed an acute improvement on the contralateral hip flexion passive ROM after a unilateral hamstring SAFR intervention. Similar crossover and nonlocal effects were also reported by others for the ankle dorsiflexion ROM (*Garcia-Gutierrez et al., 2018*; *Kelly & Beardsley, 2016*). However, in a study where a small area (the sole of the foot) was rolled, no statistically significant crossover effects were demonstrated which, according to the authors, may be the result of insufficient afferent (e.g., mechanoreceptors, nociceptors, and proprioceptors) feedback from the rolling induced-mechanical pressure on tissue (*Grabow et al., 2017*).

Recent work suggests that foam rolling can also have nonlocal effects potentially influencing muscle mechanical sensitivity. For example, high-intensity foam rolling or rolling massage may decrease the contralateral or nonlocal muscle mechanical sensitivity (*Aboodarda, Spence & Button, 2015*; *Cavanaugh et al., 2017b*; *Cheatham & Baker, 2017*; *Cheatham, Stull & Kolber, 2019*), thereby leading to an enhancement of the stretch tolerance of these muscles. Recent work from Behm's group (*Aboodarda et al., 2018*; *Cavanaugh et al., 2017a*; *Young, Spence & Behm, 2018*) showed that the intervention of foam rolling or rolling massage could even cause the reduction of excitability at the muscular, spinal, and supraspinal levels. Thus, with the decreased corticospinal (*Aboodarda et al., 2018*) and spinal (*Young, Spence & Behm, 2018*) excitabilities, it is interesting and important to

examine if there are any central-mediated changes on the contralateral or nonlocal muscles, due to the intervention of unilateral SAFR.

*Cavanaugh et al. (2017a)* and *Cavanaugh et al. (2017b)* found that three, 30-s unilateral heavy rolling massage on the calf muscle impaired the force generation during the first 200 ms (F200) but not the peak torque for the ipsilateral muscle maximal voluntary isometric contraction (MVIC). In addition, they also found that the F200 of the contralateral non-intervened muscle did not change after massaging the contralateral muscle (*Cavanaugh et al., 2017b*). It is important to mention that, during the early phase (e.g., 25–75 ms) of an explosive contraction, the RFD can be influenced by neural factors such as the maximal motor unit firing rate, the motor unit high frequency discharges (doublet) at the onset of muscular action, and the reduction in motor unit recruitment thresholds linked to central mechanisms (e.g., increased spinal motoneuron excitability) (*Maffiuletti et al., 2016*; *Rodriguez-Rosell et al., 2018*). For contraction of longer duration (>75 ms), the RFD is more influenced by speed-related properties of the muscle and MVC force per se (*Maffiuletti et al., 2016*). Thus, examining the RFD during the first 50 ms, or the first 100 ms from the onset of an explosive contraction may serve as a better and more sensitive parameter than the RFD parameters during the later phase or peak force, for monitoring potential altered excitation-induced changes in the force generating ability of the nonlocal muscles (*Maffiuletti et al., 2016*). In addition, these RFD parameters often serve as better predictors of motor/athletic performance than the peak force in real life scenarios such as an athletic event (*Maffiuletti et al., 2016*).

Therefore, the main purpose of this study was to investigate whether 10 sets of 30-s unilateral SAFR intervention could induce potential changes in the RFD parameters (the RFD of the first 50 ms: $RFD_{0-50}$; $RFD_{0-100}$; and $RFD_{0-200}$) of the contralateral hamstring muscle groups. This study is based on a project where other parts of the data was published previously (*Killen, Zelizney & Ye, 2018*). With the larger sample size from the current data pool, the effects of unilateral SAFR on the contralateral non-intervened muscle flexibility, isometric strength, as well as the prime movers' surface electromyographic (EMG) amplitude were also examined. If the contralateral $RFD_{0-50}$ and $RFD_{0-100}$ changed after foam rolling, then it is possible that neural mechanism(s) might have played a role influencing the contralateral non-intervened limb muscles. As shown in *Cavanaugh et al. (2017b)*, the $RFD_{0-200}$ and the peak force of the contralateral muscle group would not likely to be statistically different following the unilateral SAFR. This work will build upon previous work by explicating the effects of high volume, long duration SAFR on musculoskeletal functions. The potential practical applications of this research may not be suitable for athletic performance, but may be of great importance for clinical practice (*Cheatham et al., 2015*). For instance, as physical therapists may have their patients utilize relatively high-volume, long duration SAFR to improve unilateral limb ROM, it is also important to understand the potential rolling-induced effects on the non-intervened contralateral limb muscles.

## MATERIALS & METHODS

### Subjects

Thirty-four adults (male: $n = 13$, mean $\pm$ SD age = $24 \pm 4$ years; height = $174.3 \pm 9.2$ cm; body weight = $84.3 \pm 15.2$ kg; female: $n = 21$, age = $21 \pm 1$ years; height = $162.8 \pm 3.5$ cm; body weight = $65.5 \pm 13.4$ kg) participated in this investigation. All subjects were healthy and physically active (performed resistance exercises at least once per week, and aerobic exercises at least twice per week 6 months leading up to this study). In addition, all subjects were familiar with foam rolling, and they all had the experience of performing foam rolling exercises previously. Prior to any experimental testing, each subject completed an informed consent and a pre-exercise health and exercise status questionnaire, which indicated no current or recent neuromuscular or musculoskeletal disorders in the lower body. During the consenting process, the subjects were instructed to maintain their normal habits in terms of dietary intake, hydration status, and sleep during the investigation. In addition, they were refrained from performing any upper or lower body resistance exercise at least 72 h prior to each testing session. All experimental procedures for this investigation were approved by the University Institutional Review Board (Approval Code: 17-062).

### Design

This investigation used a within-subjects randomized crossover design to examine the potential crossover effects of unilateral hamstring SAFR. Specifically, dependent variables included contralateral hip flexion passive ROM, knee flexion RFD parameters ($RFD_{0-50}$, $RFD_{0-100}$, and $RFD_{0-200}$) and isometric strength, as well as the hamstring muscles biceps femoris (BF) and semitendinosus (SEMI) EMG amplitude before (Pre) and immediately after (Post) the unilateral SAFR. After the first visit served as the familiarization to practice generating explosive isometric force and the SAFR protocol, the following experimental visits were conducted with a randomized order, during which the SAFR and control conditions were delivered. Between visits, a minimum of 24 h of rest was provided. All measurements were taken from the subject's non-dominant (NONDOM) leg, and the interventions were performed on the subject's dominant (DOM) leg. Five subjects (three males) were left-footed, based on the leg dominance test (which foot the subject would kick a soccer ball).

### Procedures

Upon arrival during each experimental visit, the subjects were instructed to lie down on a medical bed in the supine position for a two-minute rest. The pre-tests were then conducted as the following order: NONDOM hip flexion passive ROM (Pre-ROM), and NONDOM knee flexor isometric strength testing (Pre-MVIC). Following the baseline tests, either the control or SAFR intervention was performed. During the control visit, the subjects lay down on the same medical bed for ten minutes. During the SAFR visit, the subjects performed the same protocol as described in *Killen, Zelizney & Ye (2018)*. Briefly, with a foam roller (FitPlus Premium High-Density Foam Roller, Fit Plus LLC, Chattanooga, TN, USA) placed between the dominant hamstring muscle group and a yoga mat, the subject was asked to foam roll the hamstring muscles with a 1-s up (roll to the ischial tuberosity)/1-s

down (roll to the popliteal fossa) tempo. During the SAFR, the DOM leg was maintained with an extended position, and the subject crossed the NONDOM ankle on top of the DOM leg. Ten sets of 30-s SAFR with 30-s rest (lie down on the yoga mat with both legs extended and relaxed) between sets were performed by the subjects. Immediately following the intervention, tests (Post-ROM and Post-MVIC) were conducted using the exact same order and manner as during the pre-measurements.

## Measurements
### Hip flexion passive ROM
The straight leg raise test was used to measure the NONDOM hip flexion passive ROM in the supine position. With a Baseline® Bubble® inclinometer (Fabrication Enterprises Inc., White Plains, NY, USA) placed on the NONDOM knee cap, a member of the research staff grasped the ankle of the DOM leg and raised it slowly to the point where the subjects first felt tension from the hamstring muscle group. The value from the inclinometer was then recorded as the NONDOM hip flexion passive ROM. To ensure measurement consistency, the researcher used a Sharpie pen to mark the location where the inclinometer was placed, and for subsequent measurements, the researcher placed the inclinometer in the exact same spot to perform this measurement. In addition, extra care was taken to ensure the NONDOM knee was kept straight during this procedure. At least three trials with 15-s rest between trials were performed to establish the NONDOM hip flexion passive ROM. If the values from any two trials differed more than two degrees, then extra trials would be conducted. The average of the three closest trials was then calculated and recorded as the NONDOM hip flexion passive ROM.

### Isometric testing (RFD and isometric strength)
Following the measurement of NONDOM hip flexion passive ROM, the subjects were instructed to lie down on the same medical bed with the prone position. With both ankles hanging off the edge of the medical bed, the subjects kept the knee joints straight and relaxed. The research staff then put a cuff around the NONDOM ankle and connected the cuff to one end of a force transducer (Model SSM-AJ-500; Interface, Scottsdale, AZ, USA), with the other end of the transducer attached to a wooden platform mounted on the floor. Before testing the NONDOM knee flexor MVC strength, the subjects were instructed to perform three isometric contractions at about 50% of the perceived maximal effort to warmup. Specifically, they were told to "squeeze as fast as possible", as they practiced during the familiarization session. The subjects then performed three, 3-s MVICs of the NONDOM knee flexors with 2 min of recovery between the contractions. During each MVIC, the research staffs provided a verbal countdown "three, two, one, pull" to the subject, with specific emphasis on "pull as fast as possible and hard", based on the instruction recommendation by *Maffiuletti et al. (2016)*. During all maximal contractions, the research staffs provided strong verbal encouragement.

The surface EMG signals were detected through two bipolar surface EMG electrodes (input impedance >$10^{15}$ $\Omega$, DE 2.1 Single Differential Surface EMG Sensor; Delsys, Inc., Natick, MA, USA; 10 mm interelectrode distance) placed on the NONDOM BF as well as the SEMI muscles, based on the electrode placement recommendations from the SENIAM

project (*Hermens et al., 1999*). The electrodes' locations were recorded and marked with a pen to ensure the electrodes were placed on the exact same spots during both experimental visits. A reference electrode (5.08 cm diameter Dermatrode HE-R; American Imex, Irvine, CA, USA) was placed over the 7th cervical vertebrae during data collection. Prior to detecting any EMG signals, all skin sites were shaved with a razor and cleansed with rubbing alcohol. In addition, all the surface EMG sensors were firmly secured to the skin with stripes of adhesive tapes.

### Force and surface EMG signal processing

During each MVIC trial, both the force and EMG signals were sampled at 20 kHz with a 16-channel Bagnoli$^{TM}$ desktop EMG system (Delsys, Inc., Natick, MA, USA), and stored in a laboratory computer (Dell XPS 8900, Round Rock, TX) for further analyses. The EMG signals were preamplified (gain = 1,000) and went through a 4th-order Butterworth filter with the bandpass set at 20–450 Hz. A custom-built LabVIEW (LabVIEW; National Instruments, Austin, TX, USA) program was used to analyze the RFD parameters, and the peak force along with its EMG amplitude. For each MVIC, the peak force output was determined from the highest mean 500-ms portion of the force plateau during the contraction of the 3-s MVIC. The isometric strength was determined by the highest peak force output among all three MVICs. The EMG amplitude was then calculated as the root-mean-square (rms) of the same 500-ms window corresponding to the peak force of the contraction. To determine the RFD during the MVICs, we first determined the force onset point at which the force signal exceeded the baseline by 2% of the baseline-to-peak value (*Andersen et al., 2010*). The RFD was then calculated as the slope of the force-time curve ($\Delta$force/$\Delta$time) derived at time intervals of 0–50 ($RFD_{0-50}$), 0–100 ($RFD_{0-100}$), and 0–200 ($RFD_{0-200}$) ms relative to the onset of the contraction (*Aagaard et al., 2002*). The RFD values from the contraction that produced the highest peak force were selected for subsequent statistical analyses.

### Statistical analyses

Test-retest reliability was calculated across the pre-values from two experimental visits by determining the intraclass correlation coefficient (ICC; relative reliability) using Model "3,1" (*Weir, 2005*). In addition, the standard error of the measurement (SEM) was calculated for measures of absolute reliability using the equation from *Hopkins (2000a)* and *Weir (2005)*: SEM = squat root (1-ICC). Lastly, the coefficient of variation (CV) was calculated as a normalized measure of the SEM using the equation: CV = (SEM/Grand mean) $\times 100$ (*Hopkins, 2000a*).

Assumptions for normality of distribution for all model (fit) residuals were checked and confirmed using the Shapiro–Wilk test. Separate two-way (time [Pre vs. Post] $\times$ intervention [Control vs. SAFR]) repeated measures ANOVAs were performed to examine potential changes of all the dependent variables before and after the different interventions. When appropriate, the follow-up tests included paired t-tests with Bonferroni corrections (if there was an interaction). All statistical tests were conducted using statistical software (IBM SPSS Statistics 25.0; IBM, Armonk, NY) with alpha set at 0.05. In addition, effect sizes

Cohen's $d$ (*Cohen, 1992*) were calculated to assess the treatment effect (Control vs. SAFR) and time effect (Pre vs. Post) for each dependent variable. The Cohen's $d$ was calculated as $(\text{Mean}_1 - \text{Mean}_2)$/pooled Standard Deviation ($\text{SD}_{\text{pooled}}$), where $\text{SD}_{\text{pooled}} = $ square root $[(\text{SD}_1^2 + \text{SD}_2^2)/2]$ , with 0.2, 0.6, and 1.2 as the thresholds for small, medium, and large effect sizes, respectively, based on Hopkins' interpretations for the magnitude of effect size (*Hopkins, 2000b*). Lastly, separate Tufte slopegraphs with the Cumming estimation plots (Fig. 1) were generated to display the complete statistical information regarding all RFD parameters before and after interventions (Control vs. SAFR) (*Ho et al., 2018*).

## RESULTS

### Test-retest reliability
Table 1 shows the ICC, SEM, and CV of measurement variables (contralateral hip flexion passive ROM, contralateral knee flexion isometric strength, $\text{RFD}_{0-50}$, $\text{RFD}_{0-100}$, $\text{RFD}_{0-200}$, and contralateral BF and SEMI muscle maximal EMG amplitude).

### NONDOM hip flexion passive ROM
The results from the two-way repeated measures ANOVA indicated that there was a statistically significant time × intervention interaction ($F(1, 33) = 27.371$, $p < 0.001$). The follow-up paired samples $t$-tests indicated that the NONDOM hip flexion passive ROM significantly increased following the SAFR (mean ± SD: Pre vs. Post = 68.3 ± 21.0 vs. 73.2 ± 23.2, $t = 6.625$, $p < 0.001$; $d = 0.22$), but not following the control (Pre vs. Post = 68.4 ± 20.7 vs. 68.2 ± 21.2, $t = 0.301$, $p = 0.383$; $d = 0.01$). In addition, the Post-ROM value were significantly higher following the SAFR than that following the control (SAFR vs. Control = 73.2 ± 23.2 vs. 68.2 ± 21.2, $t = 2.943$, $p = 0.003$; $d = 0.23$).

### NONDOM Knee flexors isometric strength and EMG amplitude
The two-way ANOVA showed neither an interaction ($F(1, 32) = 3.784$, $p = 0.061$) nor main effects (time and intervention) for the isometric strength of the NONDOM knee flexors. In addition, no interactions (BF: $F(1, 30) = 0.038$, $p = 0.847$; SEMI: $F(1, 30) = 0.162$, $p = 0.690$) as well as main effects (time and intervention) were found for the EMG amplitude of both BF and SEMI muscles.

### RFD Parameters
For the $\text{RFD}_{0-50}$, the two-way ANOVA indicated that there was a time × intervention interaction ($F(1, 31) = 5.834$, $p = 0.022$). The follow-up paired samples $t$-tests indicated that the $\text{RFD}_{0-50}$ value significantly decreased following the SAFR (Pre vs. Post = 1626 ± 1325 vs. 1125 ± 937, $t = 2.630$, $p = 0.007$; $d = 0.44$), but not following the control (Pre vs. Post = 1,756 ± 1,611 vs. 1,778 ± 1,578, $t = 0.115$, $p = 0.455$; $d = 0.01$). In addition, the $\text{RFD}_{0-50}$ was significantly lower following the SAFR than that following the control (SAFR vs. Control = 1,125 ± 937 vs. 1,778 ± 1,578, $t = 3.100$, $p = 0.002$; $d = 0.50$).

For both $\text{RFD}_{0-100}$ and $\text{RFD}_{0-200}$, the results from the two-way ANOVAs did not show any time × intervention interactions ($\text{RFD}_{0-100}$: $F(1, 31) = 3.152$, $p = 0.086$; $\text{RFD}_{0-200}$: $F(1, 31) = 0.872$, $p = 0.358$). Figure 1 displays the individual absolute change scores (delta) of the Control vs. SAFR interventions and the paired mean changes in all RFD parameters.
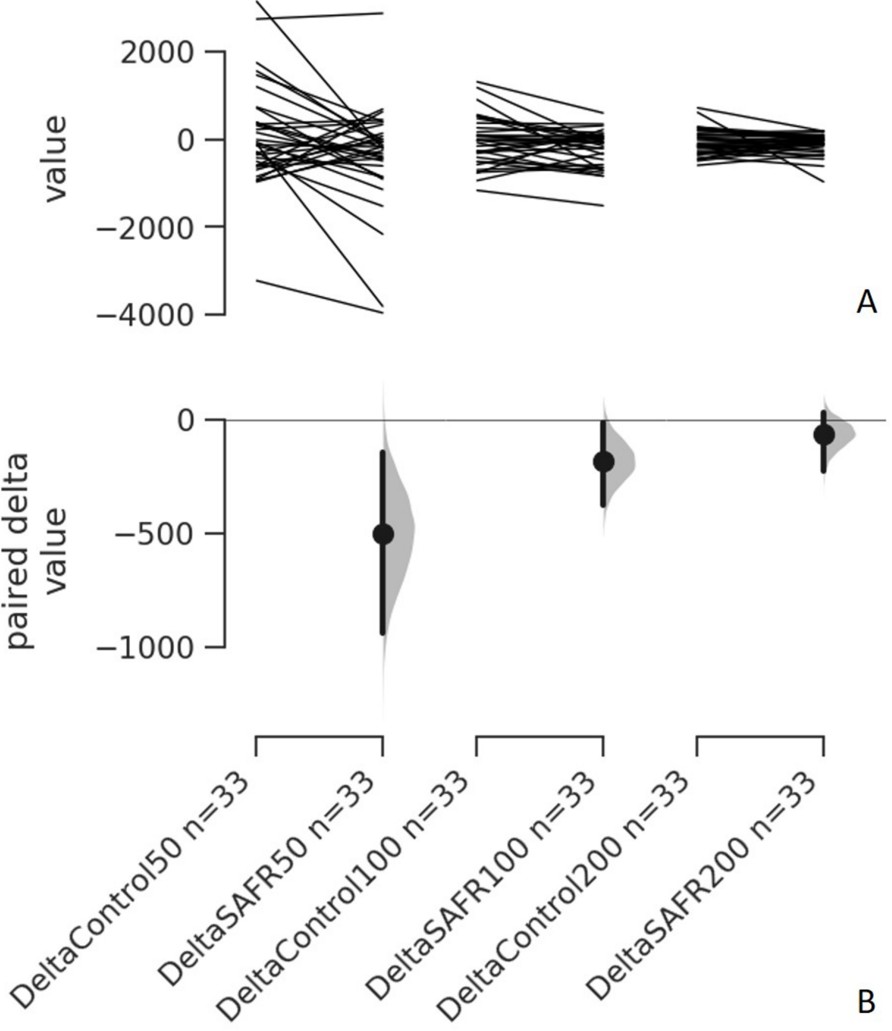

**Figure 1 Individual change responses and the paired mean differences for comparisons (the absolute change scores of the Control vs. SAFR) of all three RFD parameters (RFD$_{0-50}$, RFD$_{0-100}$, RFD$_{0-200}$).** The paired mean differences for comparisons (the absolute change scores of the Control vs. SAFR) of all three RFD parameters (RFD$_{0-50}$, RFD$_{0-100}$, RFD$_{0-200}$) are shown in the above Cumming estimation plot. The raw data (the change score) is plotted on the upper axes; each paired set of observations is connected by a line (A). On the lower axes, each paired mean difference is plotted as a bootstrap sampling distribution. Mean differences are depicted as dots; 95% confidence intervals (CIs) are indicated by the ends of the vertical error bars (B). For each paired comparison, the paired mean difference (Δ) with 95% CI are provided as the following format: Δ [95 CI: lower bound; 95 CI upper bound] DeltaControl (RFD$_{0-50}$) vs. DeltaSAFR (RFD$_{0-50}$): −503.0 [95 CI: −936.0; −149.0]; SEE = 293.89 DeltaControl (RFD$_{0-100}$) vs. DeltaSAFR (RFD$_{0-100}$): −182.0 [95 CI: 367.0; −14.5]; SEE = 125.94 DeltaControl (RFD$_{0-200}$) vs. DeltaSAFR (RFD$_{0-200}$): −64.8 [95 CI: −225.0; 29.6]; SEE = 67.99. Note: RFD = rate of force development; SAFR = self-administered foam rolling; SEE = standard error of the estimate.

**Table 1   Test-retest reliability for dependent variables.** The intraclass correlation coefficient (model "3,1") (ICC (3, 1)), standard error of measurement (SEM), and coefficient of variation (CV) of variable measurements (the contralateral hip flexion passive range of motion [ROM], the contralateral biceps femoris [BF] and semitendinosus [SEMI] EMG amplitude, the contralateral knee flexion isometric strength, and the rate of force development (RFD) for the first 50 [$RFD_{0-50}$], 100 [$RFD_{0-100}$], and 200 [$RFD_{0-200}$] ms of the maximal isometric contraction).

| Variable measures | ICC (3,1) | SEM | CV (SEM %) |
|---|---|---|---|
| Passive ROM (°) | 0.91 | 4.26 | 6.2% |
| Isometric strength (N) | 0.88 | 34.27 | 10.6% |
| EMG Amplitude of BF ($\mu$V) | 0.84 | 31.76 | 23.8% |
| EMG Amplitude of SEMI ($\mu$V) | 0.80 | 38.55 | 30.2% |
| $RFD_{0-50}$(N/s) | 0.81 | 633.00 | 37.9% |
| $RFD_{0-100}$(N/s) | 0.84 | 312.39 | 23.5% |
| $RFD_{0-200}$(N/s) | 0.87 | 184.06 | 17.2% |

## DISCUSSION

The main purpose of this investigation was to examine whether unilateral SAFR exercise could influence the contralateral muscle RFD parameters. Additionally, the contralateral limb passive ROM, and the isometric strength along with the maximal EMG amplitude were also examined. Ten, 30-s unilateral hamstring SAFR increased the contralateral passive hip flexion ROM by almost five degrees. In addition, the contralateral isometric strength along with the prime movers' maximal EMG amplitudes were not statistically significantly altered after the SAFR intervention. These results were consistent with the ones from our previous report (*Killen, Zelizney & Ye, 2018*). Regarding the RFD parameters, to our knowledge, this is the first study to investigate potential crossover effect of unilateral SAFR on the contralateral muscle RFD. Specifically, the contralateral $RFD_{0-50}$ significantly decreased following the unilateral SAFR. Relative to the control condition, the SAFR imposed a small treatment effect ($d = 0.42$) on the decrement of the $RFD_{0-50}$. In addition, the estimation plots with each paired mean difference (between deltaControl and deltaSAFR) suggest that the mean differences for deltaControl and deltaSAFR were below zero (mean deltaControl was less than mean deltaSAFR) for both $RFD_{0-100}$ and $RFD_{0-200}$ (Fig. 1).

With the $RFD_{0-50}$ demonstrating the statistically significant reduction, and the relatively larger mean differences for both the $RFD_{0-100}$ and $RFD_{0-200}$ before and after the unilateral SAFR, as compared to the control, it is also important to notice the magnitudes of the changes in these RFD parameters. Relative to the control condition, the average percent changes of the contralateral muscle RFD $_{0-50}$, $RFD_{0-100}$, and $RFD_{0-200}$ following the SAFR were $-31.2\%$, $-16.8\%$, and $-10.1\%$, respectively, showing a diminishing decline as the time interval for RFD lengthens. Folland and colleagues *Folland, Buckthorpe & Hannah (2014)* used surface EMG to examine the relative contribution from the neural and contractile components in the early and late phases of the knee extension force-time curve. Based on their multiple linear regression analysis, the agonist muscle EMG amplitude was particularly important explaining the variance in explosive force during the initial phase (e.g., 25–75 ms). Furthermore, neural factors such as the motor unit recruitment

and discharge rate, the presence of motor unit doublet discharges, the spinal motoneuron excitability, and the corticospinal excitability are also important contributing to the RFD during the initial phase of a rapid contraction (*Maffiuletti et al., 2016*; *Rodriguez-Rosell et al., 2018*). While for longer duration (e.g., >75 ms), the speed-related properties of the muscle and MVC force per se become more important for the RFD (*Maffiuletti et al., 2016*). Thus, the statistically significant reduction for the contralateral muscle $RFD_{0-50}$ might have been originated from the change(s) of one or some of the above-mentioned neural factors, induced by the unilateral SAFR.

The next obvious question is how ten, 30-s unilateral SAFR could influence the central nervous system, causing a change on the contralateral muscle RFD. In addition, another important question is where in the entire corticospinal pathway that this crossover could occur. As mentioned, repetitive high-intensity foam rolling or rolling massage exert mechanical pressure on skin, muscle, and fascia, primarily influencing mechanoreceptors and nociceptors (*Behm et al., 2013*), such as the type Ib Golgi tendon organs, the cutaneous receptors, and the type III/IV interstitial free nerve endings. A potential mechanism could be originated from the activation of the most abundant intrafascial mechanoreceptors, type III and IV receptors, due to the high-volume long duration of foam rolling. Specifically, nociceptive stimuli (e.g., rolling-induced mechanical pressure on soft tissue) could induce neurophysiological responses, including the influence on the descending pain modulation circuit (*Vigotsky & Bruhns, 2015*). In addition, diffuse noxious inhibitory control (a painful stimulus can be inhibited by another nonlocal noxious stimulus) may also serve as a central pain-modulatory mechanism influencing a nonlocal site. In the current investigation, unfortunately, it is not possible to identify the specific location that the potential crossover occurred at the corticospinal pathway, due to the lack of spinal and corticospinal excitability measurements.

Previously, *Cavanaugh et al. (2017b)* performed three, 30-s unilateral heavy (seven out of 10 based on the visual analog pain scale: uncomfortable or induced some pain with most subjects) rolling massage on the subjects' calf muscle, but only found statistically significant decrease in the RFD $_{0-200}$ for the ipsilateral, but not for the contralateral homologous muscle. An obvious difference between the current study and Cavanaugh et al. is the different interventions (foam rolling vs. rolling massage). However, review articles (*Cheatham et al., 2015*; *DeBruyne et al., 2017*) comparing foam rolling vs. rolling massage did not identify any intervention-related differences per se. Instead, the duration difference between these interventions might have played a more important role. Specifically, our subjects performed a total of 300 s foam rolls within a 10-minute period. This volume was significantly higher than the one (a total of 90 s 2-s up/2-s down rolling massage) from *Cavanaugh et al. (2017b)*). Previously, *Monteiro et al. (2017)* compared the effects of 60 vs. 120 s of SAFR on joint ROM, and found longer duration/higher volume produce larger treatment effect (*Monteiro et al., 2017*). In addition, when implementing different SAFR durations into fatiguing resistance exercise sets (e.g., three sets of 10-repetition maximum load to failure), volumes with greater than 90 s were detrimental to fatigue resistance (*Monteiro & Neto, 2016*). Thus, comparing to *Cavanaugh et al. (2017b)*, it is possible that greater amount of the nociceptive stimulus from the current high-volume,

high-intensity SAFR intervention might have induced greater inhibition, thereby showing more prominent crossover effects on the RFD parameters. In addition, the current SAFR intervention also seemed to decrease the contralateral knee flexors muscle pain perception through central pain-modulatory system, thereby leading to an enhancement of the stretch tolerance of these muscles (evidenced by the increased contralateral limb ROM).

With the novel finding of the contralateral muscle RFD parameters, we do want to point out several limitations of this investigation and emphasize a caution when interpreting the results. First, we did not specifically examine the RFD parameters of the unilateral intervened muscle group, mainly due to the primary focus of this investigation was the contralateral crossover effect. Since the crossover effects were present in the non-exercised contralateral muscle RFD parameters, the unilateral rolled muscle is more likely affected. Obviously, future studies should identify the time window of the crossover effect as well as the magnitudes of changes in RFD parameters in the unilateral intervened muscle. Besides the above-mentioned limitation, our methodology also included some limitations need to be pointed out. For example, the subjects were recruited based on convenience sampling. The investigators in this study were not blinded to the measurements of the dependent variables, which could have affected the outcome measures. In addition, the SAFR intervention should have been randomly assigned between both sides of the knee flexors, for the purpose of examining potential effect of limb dominance. Lastly, caution should be taken when interpreting the results due to the high-volume of the intervention. Specifically, it may not be appropriate to apply the findings of the study to athletic field, because most SAFR sessions prior to an exercise session or a sport event do not last more than 90 s (e.g., 3 sets of 30-s SAFR). Thus, contralateral athletic performance such as the explosive power does not necessarily decrease following a typical SAFR with a shorter duration. Instead, our results may specifically be important for areas of physical therapy and rehabilitation, as patients may undergo longer duration of interventions.

## CONCLUSIONS

A bout of ten, 30-s unilateral hamstring SAFR intervention improved the contralateral hip flexion passive ROM without altering the contralateral isometric strength performance and the prime mover muscles' maximal EMG amplitudes. However, the contralateral muscle group's ability to generate explosive force was impaired. In addition, the magnitudes of the impairments seemed to be phase-dependent (with greater impairment observed in the early phase such as the first 50 ms than those in the later ones such as the 100 and 200 ms). These results show the evidence of contralateral crossover effects of unilateral SAFR.

## ACKNOWLEDGEMENTS

The authors would like to thank all the participants who took time out of their schedules to help with this project. We also appreciate the Editor's and both reviewer' comments, which improved this manuscript significantly.

### Funding

The authors received no funding for this work.

### Competing Interests

The authors declare there are no competing interests.

### Author Contributions

- Xin Ye conceived and designed the experiments, performed the experiments, analyzed the data, contributed reagents/materials/analysis tools, prepared figures and/or tables, authored or reviewed drafts of the paper, approved the final draft.
- Benjamin S. Killen and Krista L. Zelizney conceived and designed the experiments, performed the experiments, analyzed the data, contributed reagents/materials/analysis tools, authored or reviewed drafts of the paper, approved the final draft.
- William M. Miller and Sunggun Jeon analyzed the data, contributed reagents/materials/analysis tools, prepared figures and/or tables, authored or reviewed drafts of the paper, approved the final draft.

### Human Ethics

The following information was supplied relating to ethical approvals (i.e., approving body and any reference numbers):

All experimental procedures for this investigation were approved by the University Institutional Review Board (protocol approval number: 17-062).

### Data Availability

All raw measurements are available in the Supplemental Filea.

### Supplemental Information

Supplemental information for this article can be found online at http://dx.doi.org/10.7717/peerj.7028#supplemental-information.

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
