# Peer review of "Unilateral hamstring foam rolling does not impair strength but the rate of force development of the contralateral muscle"

_PeerJ, doi:10.7717/peerj.7028_

## Round 0.1 · original submission · Major Revisions

I have now received two reviews of your manuscript. Both reviewers agree in that it needs a profound revision. Please stick to your data in your discussion and conclusion sections. You need be more clear about the number and other details in relation to the participants in the performed experiment. Please, make your experimental design clear enough to avoid any more doubts. More clarity is also needed in your statistical report. All suggestions of our reviewers should be fully considered in order to improve the quality of your work.

·

Basic reporting

The aim of the study was to compare the effect ten × 30-s SAFR vs 10 min rest on the contralateral (non-intervened) muscle’s RFD, isometric MVC and corresponding EMG. The results demonstrated a significant improvement in the hip flexion ROM and a decrease in F50 following SAFR, without any change in MVC force and EMG. The manuscript is well written and the experimental protocol is pretty clear. However, a major limitation with the study is that although simple measurement techniques (e.g. MVC force, RFD, EMG, ROM) were used, the authors have provided a lot of speculations regarding the underlying mechanisms (mainly neural factors) contributing to the results. Few of these issues have been highlighted below.
General comments
1) Introduction line 103-105: Based on what “neural mechanism” do the authors believe that SARF could alter muscle performance in the contralateral limb? This question was not elaborate in the introduction (and even in the discussion). In lines 103-105 as well as 125, the authors have provided very vague speculations about the “neural mechanisms” and ”rolling induced neural inhibition” however the reviewer could not find any evidence for this across the entire manuscript, particularly because none of the measurement techniques used in the study would directly (or even indirectly) assess these potential mechanisms.
2) Introduction line 121-125. It is not clear why the authors needed to “reexamine” the effect of SAFR on flexibility and MVC force of knee flexors. In the previous study you had 23 participants and in this experiment 34. Did you add 11 participants to your data pool, or these are 34 new participants? In either of the case, please elaborate why the reexamining the ROM and MVC force was necessary?
3) Introduction line 133-135. How could your data provide a better understanding about “potential neural inhibition …. and recovery of the immobilized limb”?
4) Methods, line 187. The examiner who measures the ROM should have been blinded to the intervention otherwise this could have affected the outcome measures.
5) Discussion, line 295-298. Although the authors found only a time effect, they decomposed the pre vs. posttest values in each intervention session and then compared the magnitude of effect sizes between the two conditions. Theoretically, the authors are NOT allowed to decompose the time effect into two separate groups. In order to do this, you need to have an interaction effect. In addition, please consider that d = 0.34 and 0.24 are already showing very small effects, so it does not make sense to compared those with 0.09 and 0.11 (while statistically there is no difference between them!).

Experimental design

Specific comments:
Abstract: The conclusion is over-reaching (see my general comment #1)
Introduction, line 108. The word “unilateral” should be replaced with “contralateral”…
Introduction, line 108-109. While Cavanaugh et al. (2017) did not find any effect on F200, it is not clear how the authors expected that F50 or F75 would result differently…
Introduction, line 112-115. This sentence needs to be revised. The structure of the sentence looks weird.
Introduction line 118: How much is the ecological validity of 10 sets × 30 s FR? This is a limitation to the results.
Methods, line 162. Is there any reason why the rolling and testing legs were not selected randomly between participants?
Methods, line 169. The order of ROM and MVC force after each intervention indicates that the MVC force of the tested leg was performed with ~ 45 s to 1 min delay, whereas in 296 you use the word “immediately” which is not correct.
Methods, line 187. The examiner should have been blinded to the protocol, otherwise this should be mentioned in the limitation.
Methods, line 189. What does slight stretch means? How much was the reproducibility of the ROM, RFD and MVC data? The ICC, CV and standard error of measurement should be reported.
Methods, line 196. Based on what criteria this 2 degrees was selected? Please provide a reference.
Methods, line 207. You used 3 contractions at 50% of MVC for warm up. What is your reference for this protocol?
Statistical analysis. Did you test the assumption of normality and sphericity for all dependent variables? Also, for the time and condition effects, you should use Bonferroni, as paired t-test is only applicable with an interaction effect.
Results line 251. There is only one figure in the manuscript (not figures)
Results line 257. It is a bit strange that the P < 0.001 but the d is very small!
Results. Overall, regardless of the p values, the effect sizes for all outcome measures (even significant ones) are very small which questions the strength of the intervention compared to the control condition.
Discussion, line 286. The increase in ROM following rolling massage is a very controversial concept, the word “as expected” indicates as if the authors have accepted the positive effect of SARF without any hesitation …
Discussion, line 295-302. Based on statistical analysis guidelines, the entire paragraph should be deleted (see my general comment # 5).
Discussion, line 309. Again, what neural factor?
Discussion, line 314. what afferent receptors? Very speculative though!
Discussion, line 316. What somatosensory training?
Discussion, line 321. Regardless of the provided explanation, that is very over-reaching, the references # 33 and 34 are both measuring the effect of static stretching, but you are using the evidence to infer the influence of SAFR. The two technique could have totally different effect.
Discussion, line 348. The stretch tolerance was not defined well, although as far as the reviewer is concerned, this is the main reason for the observed results rather than inhibitory neural effect.
Figure 1 does not elaborate anything (because it shows the pooled data). I would delete it.

Validity of the findings

The ICC, CV and standard error of measurement for RFD, MVC force and ROM should be reported.

·

Basic reporting

The authors present a manuscript on the effects of a hamstring foam rolling intervention on the range of motion, strength, and rate of force development of the contralateral limb. In general, the manuscript is clear and well-written. However, I have some suggestions that I believe will improve the quality of the paper and make some parts more transparent.

General recommendation
- When discussing statistical significance, I recommend avoiding the term "significant" in isolation (throughout), and to precede it with "statistical". This will ensure clarity in that you are talking about statistical, not clinical, significance.
- I suggest avoiding the term "muscle activity" and "activation" when referring to EMG results (see https://www.frontiersin.org/articles/10.3389/fphys.2017.00985/full)
- All discussion and reporting should be relative to the control group. I recommend not reporting inferential statistics for time or group main effects. This ignores the purpose of a control group. See comments for references.

Experimental design

No comment.

Validity of the findings

The analyses can better address the research question/design by focusing on the effects relative to the control. Discussion of the results can be more specific.

Additional comments

FIGURES AND TABLES
Figure 1 - Rather than bar plots, I recommend a plot that allows the reader to see the individual data. For example, a Gardner-Altman plot, a Cumming plot, or a slope graph. Estimationstats.com may be a good resource for inspiration.

ABSTRACT
- The abstract is subject to change given some of my other comments. I will reserve comments until all analyses are clearly relative to the control group.

INTRODUCTION
Paragraph 1 - At the authors' discretion, I suggest avoiding terms that suggest foam rolling's mechanism is tissue/mechanically-mediated (e.g., "myofascial release", "soft tissue mobilization", etc.). For instance, referring to foam rolling as a method of "massage" would be just as effective and more cautious.
L77 - I recommend avoiding anecdotal evidence. The increasing number of studies on the topic is likely enough to suggest that it's of interest for researchers and clinicians.
L88 - Not all studies have found that foam rolling does not impair performance; for example, Monteiro's work. As such, this can be reworded more cautiously, "without necessarily impairing…"
L91 - I think "heterologous" is more appropriate than "heterogeneous"
L99 - I recommend rewording this sentence to suggest that it is not the nociception in and of itself that leads to improved stretch tolerance, but potential downstream effects of nociception (e.g., descending inhibition)
L100 - "mechanical sensitivity" or "stretch tolerance" may be more precise than "pain sensitivity"
L102 - Is it the pain or discomfort that is modulating excitability or foam rolling (from a purely methods standpoint)? It is not clear to me that pain/discomfort—effects of the intervention—are causing the excitability changes. As such, I think it's more accurate to attribute these effects to foam rolling (the intervention).
L109 - I recommend explaining what is meant by "neural factors"; e.g., MU recruitment or rate coding?
L112 - More sensitive parameter than peak force for what, exactly? Altered excitation affecting the force generating ability of nonlocal muscles? Please clarify.
L115–117 - Do they always serve as better predictors? If not, it may be best to modify this sentence to read "often serve…"
L122 - "thus" is not needed
L124 - What about the semimembranosus?
L128 - Different or statistically different?
L129 - I don't think filling a gap is in and of itself important. Importance may be improving our basic knowledge of something or having clinical/real-world relevance. The following sentences start to get at this. As such, I would reword this sentence slightly. For example, "This will work build upon previous work by explicating the effects of high-volume, …"

METHODS
L139 - Why 34 subjects and the uneven split between men and women? Was it out of convenience? Were participants familiar with foam rolling?
L161–163 - Why was the intervention applied to the dominant rather than non-dominant thigh as opposed to, say, randomizing?
- Why was such a high volume of foam rolling of interest?
- For all measurements, were the observers/investigator(s) blinded at all?
- Do you have test-retest reliability information?
L196–197 - Why use the average rather than modeling the variance (e.g., using a mixed-effects model)?
L196 - How many extra trials? Why two degrees?
L223 - Do you happen to have impedance values?
L226 - 20000 Hz or 2000 Hz? The former seems high.
L229 - What type and order were the filters?
L239 - Why the linear portion from t = 0 to X rather than the dF/dt evaluated at t = X?
L246 - Are paired t-tests really of interest over the ANOVA interaction? It seems that these ignore the control group, so the results wouldn't seem very interesting.
L247 - Did you control for multiple comparisons?
L249 - How was Cohen's d calculated?
L249 - Cohen's interpretations were not intended for exercise science. Hopkins' interpretations may be preferred? https://www.sportsci.org/resource/stats/effectmag.html

RESULTS
- For all outcomes, please present the F statistic, its degrees of freedom, and the p-value.
- I suggest focusing on the ANOVA interaction (or difference between deltas) rather than main effects and post hoc testing. Inferential statistics within a condition ignore the purpose of the control group. See:
Bland, J. M., & Altman, D. G. (2011). Comparisons against baseline within randomised groups are often used and can be highly misleading. Trials, 12. doi:10.1186/1745-6215-12-264.
Bland, J. M., & Altman, D. G. (2015). Best (but oft forgotten) practices: Testing for treatment effects in randomized trials by separate analyses of changes from baseline in each group is a misleading approach. The American Journal of Clinical Nutrition, 102(5), 991–994. doi:10.3945/ajcn.115.119768.

DISCUSSION
L288 - Because joint angles are not ratio scale, you cannot have a percent change. Please remove the "7%"
L295 - This should rely on the interaction effect. If you wish to use an estimation approach, I recommend re-analyzing the data to get the estimates and their uncertainties you are interested in.
L304 - Trends do not exist within the NHST framework. As per my previous comment, if you wish to use an estimation approach, I recommend re-analyzing the data
L305–309 - Are these relative to the control group or independent of the control group?
L313–324 - I think the explanation here can be made more specific. There are different terms, lines of evidence, and hypotheses being presented, but many of them are not pointed and do not seem contiguous. For instance, there are many afferent receptors—to which ones are you referring? Reference 35 deals with reciprocal inhibition; how does this relate? I think this discussion can be made tighter.
L347–350 - Please be more specific than "neural inhibition to the central nervous system" – by what mechanism and how would this occur? It's okay to speculate, as specificity in the discussion will make it more interesting and insightful.

---

## Round 0.2 · Major Revisions

I think that the manuscript is much better now, however I would like you to incorporate the suggestions of our reviewers.

·

Basic reporting

Thank you for making the revisions.

Experimental design

No further comment

Validity of the findings

No further comment

Additional comments

There is one more comment regarding the sentence stated in line 290-293. This should be the other way round. When ANOVA shows significant main effects of times or conditions, Bonferroni post hoc comparisons (appear below ANOVA table) are reported. When an interaction effect is observed, paired t-tests with Bonferroni corrections are applied.

·

Basic reporting

no comment

Experimental design

no comment

Validity of the findings

no comment

Additional comments

I would like to thank the authors for addressing my comments. I feel that the manuscript is close, but I have a few more comments and clarifications that I feel can improve the manuscript.

INTRODUCTION
- These are small comments that I think can make the introduction more focused:
L106–109: This sentence seems opaquely speculative, and I am not sure that it adds much to the paragraph.
L110–116: These sentences are rather specific for what we know about foam rolling. It may be best to remove them and stick with the evidence that we have. E.g., "Recent work suggests that foam rolling can have nonlocal effects…"
L131–134: I think this is worth rewording or restructuring. For instance, EMG amplitude is a measure, not a mechanism, and higher level/more central mechanisms may be the mechanism by which the downstream effects (e.g., MU recruitment) occur.

METHODS
L254: I checked Delsys' website, and they do not have 20 kHz listed (https://www.delsys.com/bagnoli/). Please triple check this! If you're 100% sure, you can ignore me; I just have not seen sEMG sampled at that rate before.
L277: Please double check this equation. Shouldn't it be SD*sqrt(1-ICC)?
L280: The model (fit) residuals must be normal, not the DVs.
L290: Why 0.63 and 1.15? Shouldn't it be 0.6 and 1.2?

RESULTS
- Please include both numerator and denominator degrees of freedom for the F-tests
- I encourage the authors to remove the inferential statistics for the within-condition outcomes. This is ultimately the authors' and editor's decision, but I will reiterate that prominent biostatisticians have been discouraging within-group (i.e., time or condition) comparisons such as these. The focus should be just on the interaction effect, as this is what addresses the research question. Descriptives are appropriate for everything else:
Bland, J. M., & Altman, D. G. (2011). Comparisons against baseline within randomised groups are often used and can be highly misleading. Trials, 12. doi:10.1186/1745-6215-12-264.
Bland, J. M., & Altman, D. G. (2015). Best (but oft forgotten) practices: Testing for treatment effects in randomized trials by separate analyses of changes from baseline in each group is a misleading approach. The American Journal of Clinical Nutrition, 102(5), 991–994. doi:10.3945/ajcn.115.119768.
- Can you include the point estimate and standard error for the interaction?
- The plot looks great, but it should be for the interaction effect (difference between the deltas). The within-condition effects are not of interest.

Discussion
L346: Were not statistically significantly altered (lack of stat significance is not evidence for null; see https://www.nature.com/articles/d41586-019-00857-9)
L351: Where are you getting the Cohen's d from? Is this adjusted for the control condition?
L352–353: This sentence reads awkwardly. Please revise slightly
L354: If you update the estimation plots for the difference of differences, this sentence may change slightly. That said, I like that you're looking at your data to infer effects rather than just statistical outcomes (e.g., p-values).
L360: Are these corrected for the control condition?
L384–387: The Nijs paper deals more with chronic nociception. Consider either the reviews by Bialosky or the review by Vigotsky & Bruhns (2015) on manual therapy mechanisms. The latter may be more relevant for neurophysiological mechanisms and DNIC. The sentence will likely change slightly to accomodate the information in the cited review (e.g., the effects being in S1 vs. elsewhere).
L409: It seems uncertain if the effects are occurring in the cortices or elsewhere. It might be best to be a little bit more vague here; e.g., "have induced greater inhibition"

Conclusion
- I do not think discussion of possible mechanisms is necessary here. I would stick to what you found and can conclude from your data. You can suggest areas for future work, but I suggest avoiding heavy speculation.

---

## Round 0.3 · accepted · Accept

Thank you very much for your patience!

·

Basic reporting

no comment

Experimental design

no comment

Validity of the findings

no comment

Additional comments

I would like to thank the authors for their patience and thoroughly addressing all of my comments.